# The Autotrophic Core: An Ancient Network of 404 Reactions Converts H_2_, CO_2_, and NH_3_ into Amino Acids, Bases, and Cofactors

**DOI:** 10.3390/microorganisms9020458

**Published:** 2021-02-23

**Authors:** Jessica L. E. Wimmer, Andrey do Nascimento Vieira, Joana C. Xavier, Karl Kleinermanns, William F. Martin, Martina Preiner

**Affiliations:** 1Department of Biology, Institute for Molecular Evolution, Heinrich-Heine-University of Düsseldorf, 40225 Düsseldorf, Germany; nascima@uni-duesseldorf.de (A.d.N.V.); xavier@hhu.de (J.C.X.); bill@hhu.de (W.F.M.); preiner@hhu.de (M.P.); 2Department of Chemistry, Institute for Physical Chemistry, Heinrich-Heine-University of Düsseldorf, 40225 Düsseldorf, Germany; kkleinermanns@yahoo.de

**Keywords:** chemolithoautotrophy, early metabolism, serpentinizing systems, hydrothermal vents, origins of life

## Abstract

The metabolism of cells contains evidence reflecting the process by which they arose. Here, we have identified the ancient core of autotrophic metabolism encompassing 404 reactions that comprise the reaction network from H_2_, CO_2_, and ammonia (NH_3_) to amino acids, nucleic acid monomers, and the 19 cofactors required for their synthesis. Water is the most common reactant in the autotrophic core, indicating that the core arose in an aqueous environment. Seventy-seven core reactions involve the hydrolysis of high-energy phosphate bonds, furthermore suggesting the presence of a non-enzymatic and highly exergonic chemical reaction capable of continuously synthesizing activated phosphate bonds. CO_2_ is the most common carbon-containing compound in the core. An abundance of NADH and NADPH-dependent redox reactions in the autotrophic core, the central role of CO_2_, and the circumstance that the core’s main products are far more reduced than CO_2_ indicate that the core arose in a highly reducing environment. The chemical reactions of the autotrophic core suggest that it arose from H_2_, inorganic carbon, and NH_3_ in an aqueous environment marked by highly reducing and continuously far from equilibrium conditions. Such conditions are very similar to those found in serpentinizing hydrothermal systems.

## 1. Introduction

Biologists have traditionally linked the topic of C1 metabolism to thoughts about life’s origins. Haeckel (1902) posited that the first cells probably lived from CO_2_ [1], perhaps in a manner similar to organisms discovered by Winogradsky (1888), growing from CO_2_ with the help of electrons from inorganic donors [2]. The chemolithoautotrophic lifestyle—converting inorganic carbon into cell mass with inorganic electron donors using chemical energy instead of light—is common among modern microbes that inhabit environments similar to those on the early Earth [3]. Although microbiologists have traditionally favored the view that the first cells were anaerobic autotrophs [4,5,6,7], the electric spark experiments of Miller shifted the focus in origins literature from microbiology to nucleic acid chemistry [8]. The facile synthesis of nucleobases from cyanide condensations [9], Spiegelman’s in vitro RNA replication experiments using Qβ replicase [10], and the demonstration that RNA has catalytic activity [11] led to the concept of an RNA world [12] in which RNA molecules became synthesized by abiotic chemical means and then competed with one another for resources (activated ribonucleoside triphosphate monomers) via replication [13]. This line of thinking diverted attention away from the more challenging problem concerning the origin of living cells and toward the more tractable problem concerning the origin of nucleic acid monomers [14,15]. However, for critics, the allure of the RNA world concept has caveats, as RNA synthesis, regardless how good, does not alone solve the problem of how living cells arose [16]; *Escherichia coli* is clearly alive; RNA is clearly not.

### 1.1. Metabolism vs. Genetics?

As an alternative to the concept of an RNA world, Wächtershäuser’s theory of surface metabolism rekindled the idea of a chemolithoautotrophic start of life and brought energy into the origins debate, positing that the exergonic conversion of iron-sulfur (FeS) minerals to pyrite provided the thermodynamic drive to fuel the origin of biochemical pathways and the first autotrophic cells [17]. Wächtershäuser’s theory ignited a “genetics first vs. metabolism first” debate [18,19,20,21] that Lipmann had presciently perceived decades in advance [22]. While the theory of surface metabolism interfaced well with catalytic mechanisms in autotrophic cells, it did not interface well with energy conservation in the currency of high-energy phosphate bonds [23] nor did it offer inroads to accounting for the fundamental property of life that the living cell maintains itself in a thermodynamic state that is far from equilibrium. The discovery of deep-sea hydrothermal vents [24] and alkaline hydrothermal vents of serpentinizing systems [25] impacted the origins issue in that they harbor geochemically continuous far from equilibrium conditions that help to define the living state of cells [5].

The idea that life started from CO_2_ is appealing, but it only solves half the problem because both for life and for organic synthesis, CO_2_ requires a reductant. This is why serpentinizing hydrothermal systems are so interesting in the origins context. Serpentinization synthesizes H_2_, the main energy and electron source of chemolithoautotrophs, from protons and electrons within the Earth’s crust through the reduction of H_2_O by Fe(II) minerals. The amount of H_2_ generated by serpentinization is substantial, on the order of 16 mmol/kg in some modern systems [26], which is orders of magnitude more H_2_ than modern chemolithoautotrophs require for growth [27]. The synthesis of H_2_ during serpentinization is a continuous process that has been going on since there was water on Earth [28]. The further researchers explored the properties of serpentinizing systems, the more similarities they revealed to the life process [29], with compartmentation, energy harnessing, catalysis, and the chemical reactions of C1 compounds at vents converging on processes that comprise the core of carbon and energy metabolism in primitive autotrophic cells [30].

From a biological perspective, the genetics first vs. metabolism first debate misses the point because neither by itself is sufficient for life. Countering the genetics first view, cells are made of far more than RNA. Cells consist by dry weight of about 50% protein and 20% RNA, with DNA, lipids, cell wall, reserves, and metabolites making up the rest [31]. Most of the RNA resides in the ribosome, which synthesizes proteins in a process that consumes about 75% of the cell’s ATP investment in biosynthetic processes [32], whereby a large portion of the ATP that a cell synthesizes is not used for the synthesis of cell mass—it goes to what is called ATP spilling and maintenance energy [33,34].

Countering the metabolism first view, a handful of small molecules reacting with each other do not qualify as metabolism. A cell is a very complicated chemical system composed of more than 1000 individual partial reactions that harness energy and synthesize building blocks as well as polymers. A decisive property of metabolism is redox balance: the number of electrons that enter the cell in substrates has to be equal to the number of electrons that leave the cell in waste products plus those that remain sequestered in compounds of cell mass; otherwise, metabolism and life come to a halt [35]. Although most reactions in cells are catalyzed by enzymes, enzymes do not perform feats of magic; they just accelerate reactions that tend to occur anyway. Many core metabolic reactions of cells readily take place without enzymes [36,37,38]. The sum of the chemical reactions in the cell (metabolism) runs both the synthesis and the operation of the cellular machinery that produces progeny, harboring a new copy of instructions in DNA, hence heredity, which over generations forms the process called genetics.

Yet the main thing that cells do is neither genetics nor metabolism but energy harnessing because without energy, neither metabolism nor genetics can take place. Metabolism and genetics are merely manifestations of the actions of sustained sets of exergonic chemical reactions over generations. There is a third option in the genetics first vs. metabolism first debate, namely “energy first”, because it is hands-down obvious that energy has to come first [39], for without favorable energetics and energy release, neither genetics, metabolism, nor anything at all will take place, so says the 2nd law of thermodynamics. Cells themselves underscore that view, because the amount of ATP that a cell synthesizes always exceeds the amount of energy required for the synthesis of new cells during growth, often by about a factor of 3 [40] (the converse would violate the 2nd law), underscoring the point that life is an energy-releasing process. For a cell, staying alive means staying far from equilibrium, which it achieves by merely running the exergonic reactions that synthesize ATP: maintenance energy or ATP spilling [33,34]. In low-energy environments, where survival becomes more important than growth [41], maintenance energy becomes the main process of life.

### 1.2. Autotrophic Origins and Energy First Link C1 Metabolism to Vents

Is there an origins option that starts with energy first? Yes, and it is seated firmly in C1 metabolism and autotrophic origins. In 2021, serpentinizing systems have gone a long way to closing the gap between CO_2_ and cells. Convergent lines of evidence indicate that reactions of C1 compounds were not only the source of carbon for the first cells but also the source of energy at the origin of the first metabolic reactions. This is because in the reaction of H_2_ with CO_2_, the equilibrium lies on the side of the simple reduced carbon compounds that comprise the backbone of carbon metabolism in organisms that use the acetyl Coenzyme A (CoA) pathway of CO_2_ fixation—formate, acetate, and pyruvate. The synthesis of these acids from H_2_ and CO_2_ is exergonic under standard conditions [39], in cells that use the acetyl CoA pathway [6] and under conditions of simulated hydrothermal vents [30]. Hydrothermal vents are generally of interest in modern theories for origins [3] because they present continuously far from equilibrium conditions, with geochemically catalyzed redox reactions and gradients that could be tapped by the first cells for energy harnessing [29].

In hydrothermal systems, both modern and on the early Earth, the key to redox reactions, catalyst synthesis, and the formation of ion gradients, is molecular hydrogen, H_2_, which is generated by the spontaneous geochemical process of serpentinization [28,42,43]. During serpentinization, mineral catalysts awaruite (Ni_3_Fe) and magnetite (Fe_3_O_4_) are formed in situ in serpentinizing hydrothermal vents [44]. These minerals catalyze the synthesis of formate, acetate, and pyruvate as well as methane [30] in the laboratory from H_2_ and CO_2_ in the presence of only water and the mineral catalyst overnight at 100 °C and only 24 bar. It is likely, but not directly demonstrated, that hydrothermally formed awaruite and magnetite catalyze the synthesis of formate and methane found in the effluent of modern serpentinizing systems [45,46,47,48]. Serpentinization also renders the effluent of hydrothermal systems alkaline [48], generating the ion gradients that form at hydrothermal vents.

The synthesis of simple organics from C1 precursors in hydrothermal systems is the only known geochemical process that follows the same chemical route as a modern core pathway of carbon and energy metabolism [30,49]. In addition, modern organisms that use the acetyl CoA pathway for carbon and energy metabolism, acetogens and methanogens, exhibit a physiology that, among known life forms, is most similar to that inferred from genomic reconstructions for the last universal common ancestor of all cells, LUCA [50]. This implicates acetogens and methanogens that lack cytochromes and quinones as very primitive microbial lineages, in line with early predictions from physiology [4] and with predictions based on similarities between geochemical and biochemical reactions [36]. It is also consistent with the identification of overlapping autocatalytic networks in the metabolism of acetogens and methanogens that implicate a role for small molecule reaction systems prior to the advent of both protein and RNA [51].

## 2. Methods

### 2.1. Reaction Data Collection

Metabolic reactions were gathered and curated from the Kyoto Encyclopedia of Genes and Genomes (KEGG) reaction database [52] (version December 2020) manually. Synthesis pathways for 46 target compounds (Appendix A) were obtained and curated by hand with the help of KEGG pathways [53] and KEGG modules. The 46 target compounds comprise 20 amino acids, four ribonucleoside triphosphates, four deoxyribonucleoside triphosphates, and 18 cofactors shown in Figure 1. KEGG lacked biosynthetic pathway information on iron–sulfur clusters, so these were not included. Although depicted in Figure 1, polymers and the genetic code are also not part of the target set. The reductive acetyl CoA pathway as well as the reverse tricarboxylic acid cycle (rTCA) cycle were added to the reaction set, covering the basal CO_2_ fixation along with the gluconeogenesis and pentose phosphate cycle, allowing for the synthesis of key intermediates needed to produce amino acids, nucleic acids, and cofactors from α-ketoacids, sugars, and aldehydes. Nitrogen fixation pathways were not included, since NH_3_ gets incorporated via amino acid synthesis. If a pathway was unavailable in KEGG, it was manually reconstructed based on KEGG pathway maps. The collection of reactions unfolds in a short example: For methanofuran biosynthesis, KEGG module M00935 was used to add reactions R10935, R11038, R11039, R00736, R10902, and R11040. The very last step of producing methanofuran is missing in the module. This reaction from APMF-Glu is depicted in pathway map00680; thus, it was added manually.

In all pathways collected, oxygen-dependent reactions were either replaced with an anaerobic alternative if possible or omitted if not. This was the case for the synthesis of dimethylbenzimidazole, which is a precursor for cobamide. Although an anaerobic synthesis pathway for this precursor is known, starting from 5-aminoimidazole ribotide (short AIR) [54], several other intermediaries are not implemented in KEGG yet. Neither was there an anaerobic alternative for the production of 2-phospholactate as a precursor in the F_420_ synthesis pathway available. For both precursors, dimethylbenzimidazole and 2-phospholactate, as well as reduced ferredoxin (involved in the reductive acetyl CoA pathway) and reduced flavodoxin (in the rTCA cycle), we assume them to be producible in an unknown way in early metabolism. Assuming the reactions in question arose before the genetic code, the according proteins were presumably replaced by an alternative at that early period. Three reactions were constructed manually, because they appear as a dashed line in KEGG pathways with no corresponding reaction identification number. The reactions named *RMAN1-3* are presumed to be incomplete, since only the key compounds were listed. Two reactions are affected within tetrahydromethanopterin synthesis and the very last step was within methanofuran synthesis. Involved chemical elements such as molybdenum, sulfur donors, cobalt, and nickel were assumed to be present in the environment. During curation of the final reaction set, redundant reactions occurring in multiple syntheses were reduced to a single occurrence, such as the reaction chorismate <=> prephenate that occurs in both tyrosine and phenylalanine syntheses.

For the detection of autocatalytic cycles within cofactor biosynthetic pathways, catalysis rules (indicating which cofactors are used as catalysts in each reaction) were gathered from [51] (Appendix A) [51]. Autocatalysis is assumed if a target is needed as a catalyst within its own biosynthetic pathway.

### 2.2. Visualization of the Autotrophic Core Network

An undirected metabolic network showing the autotrophic core was generated in simple interaction format (sif) using a custom Python script. The resulting network consists of the given 404 metabolic reactions and 380 involved compounds. The bipartite network was visualized using CytoScape [55] v. 3.8.0. One partition class corresponds to reaction nodes (diamonds), the other one corresponds to compound nodes (circle-shaped). The latter were sized according to their node degree. Target compounds were colored in blue, whereas reaction nodes are depicted smaller and in gray.

### 2.3. Different Core Reaction Sets Based on Distinct Identification Approaches

Two additional reaction datasets were used to determine their intersection with the 404 reactions of the autotrophic core. The LUCA set, containing 355 genes, was identified via the phylogenetic approach [50], translating to 163 metabolic reactions and the ancient ‘reflexively autocatalytic food-generated’ (RAF) set with 172 reactions (from [51] Figure 4). The intersection between the autotrophic core, LUCA, and the ancient RAF was determined by examining which reactions overlap in the respective analyzed datasets. In addition, the overlap of reactions between all three datasets was determined. The intersection for each comparison Appendix A.

### 2.4. Statistical Analysis

A contingency table for each highly connected compound (Table 1) was built, comparing the compound frequency in the autotrophic core with the frequency in the global prokaryote anaerobic network consisting of 5994 reactions (from [51] S1A). A significant enrichment of compound frequency in the autotrophic core compared to the global prokaryotic set was observed for *p*-values smaller than 0.05. One-tailed Fisher tests were performed using the package scipy.stats in Python 3.6 (Appendix A).

## 3. Results

### 3.1. The Autotrophic Core of Biosynthesis Requires 19 Cofactors

For the purpose of this paper, let us assume for the sake of argument that life really did start from exergonic reactions of H_2_ and CO_2_ along the acetyl CoA pathway. Why do we assume the acetyl CoA pathway as the starting point of CO_2_ fixation? Among the six known pathways of CO_2_ fixation [6,56,57], it is the only one that occurs in both bacteria and archaea, the only one that traces in part to LUCA [50], and it is the only one that has been shown in the laboratory to produce acetate and pyruvate from H_2_ and CO_2_ without enzymes, using only hydrothermal minerals as catalysts [30]. In that sense, it is the obvious choice as the starting point for metabolic evolution investigations based upon current laboratory evidence. The horseshoe (incomplete) rTCA cycle follows in Figure 1 because it is the pathway that autotrophs using the acetyl CoA pathway employ to generate C4 and C5 precursors for amino acid and other syntheses [6,36,58]. Although the incomplete horseshoe the rTCA cycle occurs in bacteria and archaea, it is fed by the acetyl CoA pathway, which is the only pathway of CO_2_ fixation that is known to occur in bacteria and archaea. The other five are known to operate in only one domain [6,56]. The rTCA cycle is also an ancient pathway [59,60], and most of its reactions also operate in the laboratory in the absence of enzymes provided that pyruvate and glyoxylate are supplied as starting material, but the non-enzymatic reaction sequence operates in the oxidative direction, that is, in the absence of H_2_ and CO_2_ [37]. The acetyl CoA pathway, the rTCA cycle, and the dicarboxylate/4-hydroxybutyrate cycle, which is a derivative of the rTCA cycle and occurs only in archaea, employ O_2_ sensitive enzymes, an ancient trait [6]. The other three CO_2_ fixation pathways have no O_2_-sensitive enzymes and are typically found in aerobes, occur in only one domain each, and they arose more recently in evolution, using enzymes co-opted from preexisting pathways [6].

We also assume that the first living things on the path to cells we recognize today required the universal amino acids and bases of life, the modern synthesis of which requires in turn cofactors as catalysts. We asked: How big, exactly, is the set of reactions required for the synthesis of the building blocks of cells and the cofactors needed to make them? This gives us an impression of how challenging it would be to generate the main compounds of life at origins, with or without enzymes. We started by sketching out Figure 1a, in which the main pathways of biosynthesis in anaerobes and the amounts of main biosynthetic end products are summarized. Cofactors are usually not present in amounts that would contribute appreciably to cell mass, but they are required as catalysts. Along the acetyl CoA pathway, there are differences in the methanogenic and acetogenic versions [61].

If we look at the cofactors required to get from H_2_ and CO_2_ to pyruvate in acetogens and methanogens [6,62], we find that methanofuran, NAD(P)H, corrins, coenzyme A, thiamine, flavins, F_420_, three pterins—folate, methanopterin, and the molybdenum cofactor MoCo—as well as FeS clusters, a prosthetic group of proteins that we count as a cofactor here, are required. That is a substantial cofactor requirement, not to mention the enzymes that hold those cofactors in place for function in the pathway. The mere requirement for those 11 complicated organic cofactors would appear to make the proposition that C1 metabolism from CO_2_ to pyruvate could be the first pathway [36] seem downright absurd, were it not for the recent observation that a bit of metal, awaruite (Ni_3_Fe), or a piece of iron oxide magnetite (Fe_3_O_4_), can also catalyze the synthesis of pyruvate from H_2_ and CO_2_ [30] overnight at 100 °C in water. As a set of chemical reactions, the acetyl CoA pathway is older than the genes that encode its enzymes [58], and it is also older than the cofactors required by those enzymes.

By the foregoing count, it takes 11 cofactors to synthesize pyruvate in the modern pathway, whereby we have not counted the two steps requiring pyruvoyl enzymes at decarboxylation steps in the CoA (pantothenate) synthesis pathway; the pyruvoyl cofactor is synthesized from serine residues in the polypeptide chain of the enzyme [63]. Very surprisingly, only three additional cofactors (biotin, pyridoxal phosphate, and SAM) are required for the synthesis of the 11 other cofactors plus the main nucleosides of nucleic acids and the 20 amino acids, whereby only two more (coenzyme M and coenzyme B) are required specifically in the methanogenic pathway of energy conservation. That makes a total of 19 cofactors (counting NAD and NADP separately as well as the flavins flavin mononucleotide (FMN) and flavin adenine dinucleotide (FAD) along with two corrins F_420_ and cobamide) to support their own synthesis plus the synthesis of four ribonucleoside triphosphates, four deoxyribonucleoside triphosphates, and 20 amino acids. The genetic code and polymers are not included in the autotrophic core. In total, that makes a list of 47 target compounds (19 cofactors, 8 nucleotides, and 20 amino acids; Appendix A) that would be required to synthesize the substance of cells, as summarized in Figure 1.

The starting point of Figure 1a is H_2_ and CO_2_. Critics of autotrophic origins will be quick to point out that cyanide chemistry can readily give rise to amino acids and bases under laboratory conditions [14,64], such that we need not worry about Figure 1. However, in reply, we would be equally quick to point out that there are 415 distinct reactions in microbial metabolism involving CO_2_ as a substrate in either the forward or reverse direction [65], but there are no reactions known to us in which cyanide serves as a main source of carbon in core anabolic metabolism. Some bacteria can convert CN^–^ to CO_2_ and NH_3_ or formate and NH_3_ for growth [66,67], because CO_2_, formate, and NH_3_ readily enter metabolism, whereas cyanide does not. In other words, CO_2_ directly enters and exits the organic chemistry of the cell substance at 415 reactions, where cyanide puts up a zero. We interpret the fact that cyanide has nothing to do with modern metabolism as a clear metabolic fossil: no role for cyanide in modern metabolism indicates that cyanide had nothing to do with primordial metabolism either, or was at best <1/415th as important as CO_2_. In that sense, the main message of Figure 1 is the overall scheme, the metabolism of cells, not that it contains amino acids and bases as products.

That brings us to nitrogen. If carbon did not enter metabolism via cyanide, then the same must be true for nitrogen. If not via cyanide, how did N enter metabolism? All the amino acids, bases, and cofactors contain nitrogen (except coenzyme M). Nitrogen enters metabolism as NH_3_ (NH_4_^+^ is very unreactive) with N atoms replacing O atoms in amino acids, either via an acyl phosphate intermediate in the glutamine synthase reaction or via reductive aminations of 2-oxoacids [36,68]. An exception is the carbamoyl phosphate synthase reaction, in which NH_3_ reacts with carboxyphosphate to form carbamate in pyrimidine and arginine biosynthesis. Of course, NH_3_ is synthesized from N_2_ by nitrogenase to make it available for incorporation into organic compounds. However, N does not enter metabolism as N_2_; it enters metabolism as NH_3_, which is why we selected NH_3_ as the source of nitrogen in Figure 1. Similarly, sulfur enters metabolism as H_2_S in cysteine synthesis from serine, either via serine activation as *O*-acetylserine or *O*-phosphoserine [69]. N and S enter metabolism as dissolved gasses (NH_3_ and H_2_S) via amino acid synthesis [36]. In cells that live from H_2_ and CO_2_, C, N, S, and electrons (H_2_) enter metabolism as gasses.

### 3.2. Enzymatic Reactions in the Autotrophic Core

Figure 1a depicts the relationships among reactions that underpin the core synthesis of cells from H_2_, CO_2_, and NH_3_, but it does not depict the reactions themselves. To find out which, what kind of, and how many reactions are required to synthesize 18 cofactors, 8 nucleotides, and 20 amino acids from H_2_, CO_2_, NH_3_, and H_2_S, we turned to KEGG pathways using Figure 1 as a framework to identify the reactions and enzymes that catalyze them. The metabolic network for the 404 reactions (Appendix A) that comprise the autotrophic core is shown in Figure 2.

Other than supplying a greater level of detail than Figure 1, and showing the relative size of nodes, the network itself in Figure 2 is not hugely informative, but some of its properties are. Keeping in mind that Figure 2 comprises the marrow of modern metabolism, hence reactions that were present in life’s common ancestor, we first asked what the most highly connected metabolites are. The fifteen most common metabolites are given in Table 1. The most common compound in the autotrophic core is by far H_2_O. As stated above, water is the solvent of life’s chemistry and its most common reaction partner. Proponents of the RNA world generally view water as a poison for origins, because it promotes the hydrolysis of RNA [70]. However, the host rocks of serpentinizing hydrothermal systems are replete with environments of low water activity, mainly because water is consumed by rock in the serpentinization process [71,72]. Furthermore, fluctuating water activities that occur during serpentinization can be conducive to polymerization reactions [71]. Life counteracts the hydrolysis problem by coupling nucleic acid and protein polymerization reactions to exergonic reactions via ATP synthesis and hydrolysis such that polymer synthesis vastly outpaces hydrolysis [58]. Accordingly, ATP is the second most common reactant in the autotrophic core (Table 1), right before protons. Protons are of course normally bound to water as H_3_O^+^, although they are not counted as water here. Protons arise in hydride transfer reactions involving NADH and NADPH which yield NAD^+^ and NADP^+^, respectively. The frequency of protons in the network mainly reflects the frequency of NAD(P)H-dependent redox reactions in the autotrophic core (Table 1).

Among reactions that involve the formation or alteration of bonds with carbon atoms, the most common carbon-containing compound in the autotrophic core is, fittingly, CO_2_, which underscores the CO_2_-dependent nature of core metabolism. Among the 404 reactions in the core, 49, or every eighth reaction, involves CO_2_. This can be seen as physiological evidence in favor of autotrophic origins. The next most common carbon backbone in the core is glutamate, which is the main workhorse of nitrogen metabolism. Glutamate arises as a product in amidotransferase reactions involving glutamine as an amino donor and in transamination reactions that produce 2-oxoglutarate, which is also among the top 15 reactants in the core. ATP hydrolysis products P_i_ and PP_i_ round out the list as well as pyruvate, which links the acetyl CoA pathway to sugar synthesis and the reverse citric acid cycle [73] and is a common starting point for cofactor synthesis in the autotrophic core (Figure 2). Last among the top fifteen is NH_3_, which is often donated to biosynthetic reactions from glutamine via an amidotransferase [74] during the enzymatic reaction, without being released as free NH_3_ in the cytosol.

We identified five autocatalytic cycles in the network, that is, cofactors that are required for their own biosynthesis: pyridoxal phosphate and thiamine, whose biosyntheses were previously identified as autocatalytic cycles [36], plus ATP, NAD, and NADP. Though not contained within our set, Davidson recently reported that coenzyme A is required for activation of the complex that synthesizes the active moiety of decarboxylating pyruvoyl enzymes, which are involved in CoA biosynthesis [63]. That would make a sixth autocatalytic cycle.

### 3.3. Comparison of the Autotrophic Core with LUCA’s Genes and Ancient Autocatalytic Sets

Other recent papers have addressed the nature of ancient metabolism by looking at phosphate-independent reactions among all KEGG reactions [75], the properties of thioester-dependent reactions [76] or chemical investigations of metabolic reactions without enzymes [30,37,49,77]. A different approach has been to focus on evidence for the nature of ancient microbial metabolism that is recorded in the genomes and metabolism of bacteria and archaea. A phylogenetic approach to ancient microbial metabolism uncovered 355 genes present in bacteria and archaea trace to LUCA on the basis of vertical intradomain inheritance as opposed to archaeal–bacteria transfer [50]. Autocatalytic networks called RAFs, for reflexively autocatalytic food-generated networks, have been identified in the metabolism of anaerobic autotrophs, with an ancient RAF of 172 genes that overlaps in the metabolism of H_2_-dependent acetogens and methanogens [51]. Do these sets overlap with the autotrophic core, and if so, how?

A comparison of these three sets (Appendix A) reveals that among the 404 reactions of the autotrophic core, only 24 are represented among the 355 genes (6%) that trace to LUCA. That low degree of overlap is not surprising for two reasons. First, only a fraction of genes that trace to LUCA by phylogenetic criteria were involved in amino acid or cofactor biosynthesis, most being involved in ribosomal biogenesis or other categories. Second, only 3% of all genes shared by bacteria and archaea were not subjected to bacterial–archaeal transfers by the measure of phylogenetic trees [50], which is a criterion that played no role in the construction of Figure 1. However, it is very noteworthy that all of the cofactors shown in Figure 1, with the exception of the archaeal-specific cofactors CoM and CoB, do trace to LUCA via phylogeny, because enzymes that trace to LUCA possessed those cofactor requirements for activity [50]. In that sense, there is excellent agreement between the physiology of LUCA as inferred from phylogeny and the present autotrophic core, their commonality being cofactors, organic catalysts that are smaller and involved in a greater number of reactions than any individual enzyme.

Among the 172 reactions present in the ancient autocatalytic network shared by acetogen and methanogen RAFs [51], 81 (47%) are present in the autotrophic core. This substantial overlap also makes sense, because all cells use the same amino acids and because both this study and Xavier et al. [51] focused on bacteria and archaea that use the acetyl CoA pathway, which by itself involves almost all of the cofactors shown in Figure 1 as it operates in bacteria and archaea. That is again noteworthy, because even though pyruvate, the central C3 product of the acetyl CoA pathway [6], can be obtained from H_2_ and CO_2_ using only simple minerals as catalysts [30], the biological pathway requires about a dozen enzymes and cofactors. These cofactors trace to LUCA [50], are well represented in RAFs [51], and comprise the basal foundation of the ancient autotrophic core (Figure 1b). Clearly, in early metabolism, cofactors and the catalytic minerals that were their inorganic precursors were very important [78]. Although self-evident, this indicates that there existed a vectorial progression in metabolic evolution that centered around the nature of catalysts: from transition metal minerals to organic cofactors to enzymes, each adding specificity and rate enhancement to exergonic reactions that tend to occur anyway. The retention of transition metal centers in some enzymes, such as carbon monoxide dehydrogenase, acetyl CoA synthetase, hydrogenases, or nitrogenase, suggests that microbes have been unable to invent catalysts that can perform the same reactions without the help of electrons in the *d*-orbitals of transition metals.

The comparison with 5994 anaerobic prokaryotic reactions (see S1A in [51]) tells us which compounds are enriched in the autotrophic core. Appendix A shows that this is true for ATP (and ADP plus P_i_), CO_2_, glutamate, pyruvate, and 2-oxoglutarate. This suggests a more crucial role of these compounds in the origin of the core subsequent to later evolution in anaerobes, reflecting a process of carbon backbone elongation from CO_2_ at the heart of the core as a supply of precursors for cofactor and amino acid biosynthesis, the latter being the starting point for nucleotide biosynthesis [78].

## 4. Conclusions

It is human nature to wonder about the origin of life, which is an issue that is among the most debated of all scientific questions. However, in comparison to questions concerning the existence of dark matter or how consciousness works, the origins process lies in the ancient past, and its events are only accessible through inference. Debates within the origins field can be fierce and have a long history. They hinge upon definitions about what qualifies as being alive, what one assumes to be the habitat that brought forth the first biochemical reactions, what came first, small molecule metabolism and proteins or nucleic acids and genetics, what the nature of first energy source(s) was that the early life forms harnessed in order to grow, and what kinds of chemical compounds existed before the first energy-releasing reactions germane to modern metabolism started taking place [21]. The literature harboring those debates is generally exhausting, because the same arguments resurface over and over again. The more broadly one reads the literature on early evolution, the more one gets the impression that scientists not only do not agree about origins and the nature of the first forms of life, but worse, that scientists know little about early evolution, leaving the topic open to unconstrained speculation and argument. That puts the origins field at risk of defining scientific progress in the units of debate preparation and presentation skills rather than units of empirical findings that are linked to the explanandum (real life); it also risks vulnerability to criticisms about the role of dogma in science.

Biologists tend to hold that there are traces of early evolution preserved in metabolism itself [4,6]. While there is no obvious proof for that conjecture, the nature of basic building blocks of life is dramatically well conserved across all cells [79]. All life forms we know use proteins made of amino acids, nucleic acids made of purines, pyrimidines, sugars, and phosphate. That means that the first forms of life from which all modern forms descend had that core chemistry in place, in addition to the universal genetic code to transfer information from nucleic acids to protein at the ribosome. This adds direly needed constraints to the origins problem. By looking at metabolism from a comparative standpoint, one can distill insights into the nature of early cells.

Here, we have identified 404 reactions that comprise the autotrophic core. It contains five small autocatalytic cycles in which cofactors participate in their own synthesis. The core represents a collection of reactions that underpin the synthesis of RNA and proteins. It was present in the first cells, but it can hardly have arisen all at once. The aqueous synthesis of pyruvate from H_2_ and CO_2_ using only solid-state metal or metal oxides as a catalyst [30] indicates that the core itself likely started from H_2_ and CO_2_ and grew outwards from pyruvate while incorporating nitrogen from NH_3_. How complex the core could have become prior to the origin of enzymes is a question for future study. However, let us keep in mind that enzymes just accelerate reactions that tend to occur anyway. It is well known that many enzymatic reactions take place without enzymes [36], although sometimes, the non-enzymatic reaction rates can be so slow as to be irrelevant [80]. However, it was also demonstrated that citric acid cycle reactions [49,81] and a number of reactions involving sugars in central metabolism [77,82] can be catalyzed non-enzymatically. This suggests that a fairly complex system of reactions, yet with far less specificity than that in the core, could have arisen before the advent of genes and proteins.

H_2_O is the most common reactant in the autotrophic core, indicating an aqueous environment during its formation. That environment was not only aqueous but also reducing, as revealed by the abundance of redox reactions in the autotrophic core, the central role of CO_2_, and the circumstance that the core’s main products (amino acids and nucleic acids) are far more reduced than CO_2_. Furthermore, the number of central reactions depending upon the hydrolysis of high-energy phosphate bonds indicates that the core arose in the presence of a continuous and highly exergonic chemical reaction capable of continuously synthesizing high-energy phosphate bonds, both before and after the origin of enzymes; here, an H_2_-dependent CO_2_ reduction to acetate [30] forming acyl phosphate bonds [58] is the proposition.

Thus, the chemical reactions of the autotrophic core suggest that it formed in an aqueous environment that supplied H_2_, CO_2_, and NH_3_, was highly reducing, and harboring continuously far from equilibrium conditions. Those conditions are very similar to those found in serpentinizing hydrothermal systems [44,77], and furthermore, they are very similar to those inferred from the functions of enzymes that vertically trace to the last universal common ancestor [50,83].

Notwithstanding pyrrolysine [84], selenocysteine [85], and a number of modified bases [86], the lack of fundamental deviation among modern life forms from the core building blocks of life, core information processing, and the core repertoire of cofactors [87] indicates that whatever chemical processes occurred at origin did not give rise to alternative cores with enough staying power to persist to the present. “Still, other cores could have existed” the critic might interject, which is true. “But even if they existed, they are irrelevant”, we would counter, because they are disjunct from the biologist’s explanandum: the autotrophic core that we can observe in modern life forms.

## Figures and Tables

**Figure 1 microorganisms-09-00458-f001:**
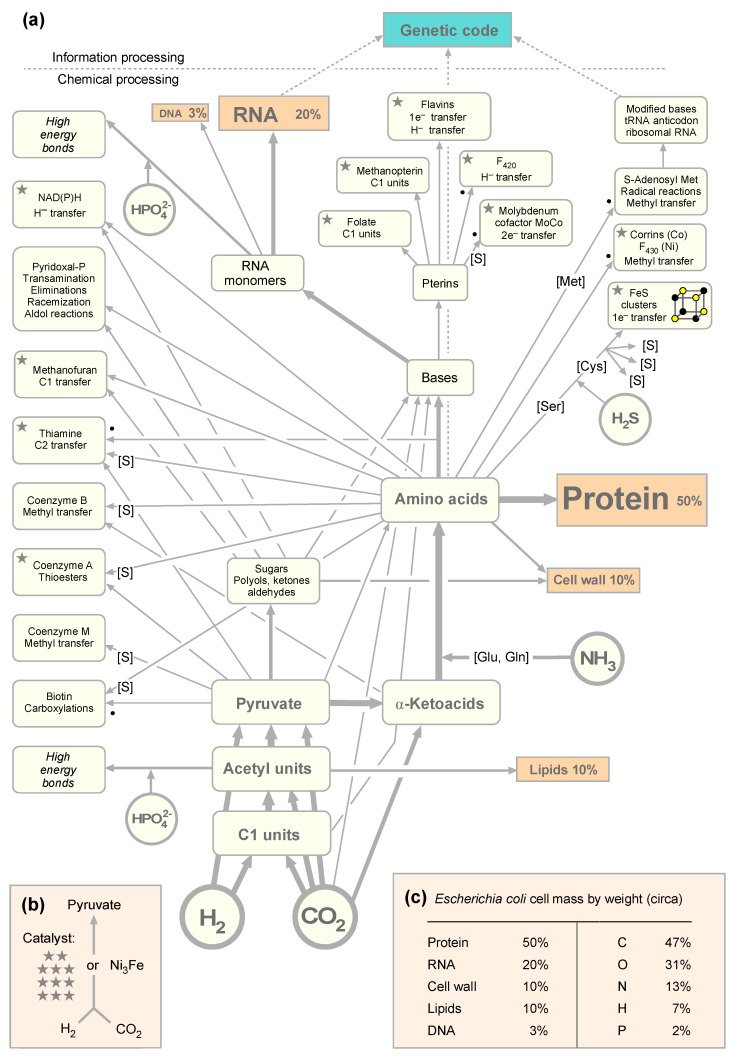
(**a**) A general map of core metabolism. The arrows in the map do not cover every atom in every cofactor, amino acid, or base, showing main mass contributions instead. A dot indicates that radical S-adenosyl methionine (SAM) enzymes are involved in the biosynthetic pathway leading to the product. [S] indicates that sulfur is incorporated in the biosynthetic pathway. (**b**) Cofactors indicated by a star are required in the pathway from H_2_ and CO_2_ to pyruvate in either acetogens or methanogens or both. (**c**) The composition of cells in terms of its main components and elemental contributions to dry weight (from [31]).

**Figure 2 microorganisms-09-00458-f002:**
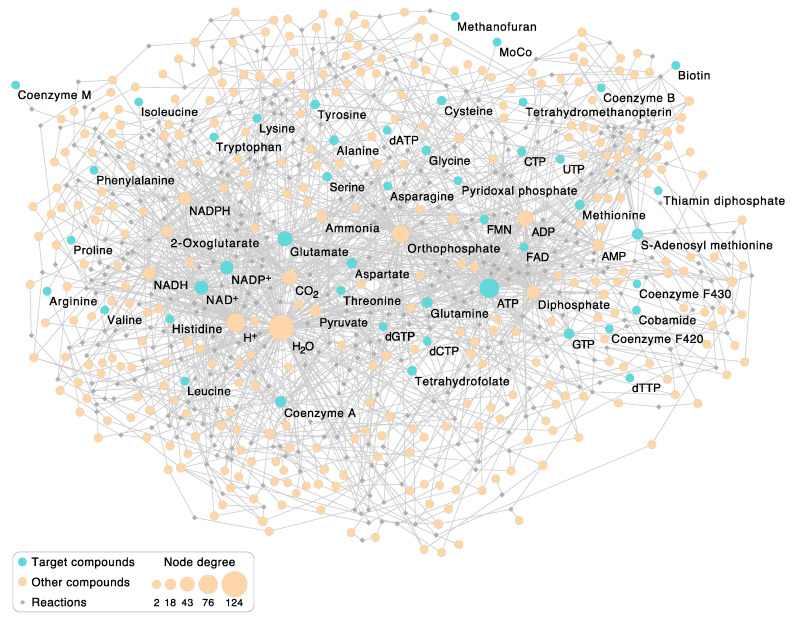
The autotrophic core network of 404 reactions underlying Figure 1. The undirected bipartite graph comprises 404 reaction nodes (displayed as gray diamonds) and 380 compound nodes (circles). The 46 target compounds are colored blue; other compounds involved in the reactions appear orange. Target compounds correspond to the core compounds in Figure 1. Each compound participating in a reaction is connected to the respective reaction node with an edge. Compounds are sized according to node degree (number of reactions the compound takes place in). For example, H_2_O appears either as reactant or product in 125 reactions and is the most frequent compound in the 404 reactions (see also Table 1). In primordial metabolic processes, before the existence of enzymes, a more limited spectrum of compounds than those in Figure 1 was provided by the environment. Compound nodes are labeled if they are targets or if the node degree is ≥20. Note that FeS clusters are not included in this figure since their synthesis cannot be reconstructed using KEGG. The network contains only l-amino acids.

**Table 1 microorganisms-09-00458-t001:** Highly connected nodes.

Compound	Frequency
H_2_O	125
ATP	77
H^+^	76
P_i_	66
ADP	55
CO_2_	49
Glutamate	44
PP_i_	37
NAD^+^	37
NADP^+^	35
NADPH	34
NADH	33
2-Oxoglutarate	24
Pyruvate	22
NH_3_	21

## Data Availability

Metabolic data that supports the findings of this study is available in KEGG [52]. The data presented in this study are available in Appendix A.

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
