# Peer review of "The Autotrophic Core: An Ancient Network of 404 Reactions Converts H2, CO2, and NH3 into Amino Acids, Bases, and Cofactors"

_microorganisms, 2021, doi:10.3390/microorganisms9020458_

Round 1
Reviewer 1 Report
Wimmer et. al construct a model of core metabolic pathways with the addition of proposed ancient carbon fixation pathways (specifically the Wood–Ljungdahl (WL) pathway and the rTCA cycle), and suggest this metabolic network represents ancient autotrophic metabolic core metabolism. The authors analyze this network by looking at highly connected metabolites, and show that water, CO2, small organics and energy transducing coenzymes (e.g. NAD(P)H and ATP) are highly connected nodes in their network. Interestingly, the authors note that several complex coenzymes are required for synthesizing key metabolic end products, and that some coenzymes are autocatalytic.
Overall, this paper provides a valuable resource for future studies of ancient metabolism, and thus warrants publication. That being said, the analysis provided in this paper falls short of convincing this reviewer that network properties, like metabolite connectivity, provide evidence that the network is an accurate representation of ancient autotrophic metabolism. Thus, some additional analysis and controls would help strengthen the results and interpretation presented in the paper.
Major comments:
Methods section is not entirely clear.
The authors should elaborate much more in section 2.1, where they describe how KEGG reactions were acquired, and how KEGG pathways were used to obtain the collection of reactions used in their model. The authors use the term “pathway module,” but the KEGG database has both a “pathway” and “module” database that include KEGG reactions. It would be very helpful for the authors to clarify what exactly was done in order to generate their network. Additionally, providing source code (if any) would be very helpful.
Including some more details in the SI tables.
In the SI table 2, the authors provide a list of reactions that are incorporated into the network, the authors should note which KEGG reactions were added manually to the network. As of now, it is not clear whether a reaction was added via a pathway the authors included, or via manual inclusion (for instance, because it was an anaerobic equivalent reaction for a particular pathway).
Networks under alternative assumptions
The authors rely on network properties (e.g. connectivity of different metabolites) to corroborate claims that this network is an accurate model for ancient autotrophic metabolism (for instance see L339-341). How would the network properties change under alternative assumptions? For instance, two helpful controls would be to (1) compute network properties using different carbon fixation pathways (e.g. 3-hydroxypropionate bi-cycle, the hydroxypropionate–hydroxybutyrate cycle or the dicarboxylate–hydroxybutyrate cycle), and (2) build networks from random sets of 404 anaerobic reactions in KEGG, and see if key metabolite connectivity measures (e.g. CO2/water connectivity) are still observed.
How does this network compare to reactions in LUCA?
Some of the authors on this paper previously reconstructed a model for LUCA (Weiss et al. 2016). It would be very interesting to see the overlap of reactions in this network with reactions identified to be present in LUCA. Differences between the two would be extremely interesting to highlight, and would showcase which reactions are missing from LUCA, and thus require deeper search for non-enzymatic or “latent” biochemical synthetic routes.
Minor comments:
L 89. What is a “partial reaction”?
The paper could use a brief summary of principle findings at the end of the introduction to help guide the reader into the method and results section.
There are some other papers that attempted to reconstruct ancient autotrophic metabolic networks (Srinivasan and Morowitz 2009; Goldford et al. 2019), and it might be worth referencing and highlighting key differences in approach if the authors see fit.
References:
Goldford, Joshua E., Hyman Hartman, Robert Marsland 3rd, and Daniel Segrè. 2019. “Environmental Boundary Conditions for the Origin of Life Converge to an Organo-Sulfur Metabolism.” Nature Ecology & Evolution 3 (12): 1715–24.
Srinivasan, Vijayasarathy, and Harold J. Morowitz. 2009. “The Canonical Network of Autotrophic Intermediary Metabolism: Minimal Metabolome of a Reductive Chemoautotroph.” The Biological Bulletin 216 (2): 126–30.
Weiss, Madeline C., Filipa L. Sousa, Natalia Mrnjavac, Sinje Neukirchen, Mayo Roettger, Shijulal Nelson-Sathi, and William F. Martin. 2016. “The Physiology and Habitat of the Last Universal Common Ancestor.” Nature Microbiology 1 (July): 16116.
Reviewer 2 Report
Notes on the manuscript:
276-277 unclear sentence here, please clarify your meaning
323 Exactly how does a serpentinizing geo-surround solve the water problem with regard to RNA world?-- the text here seems to quickly dismiss this potentially critical stumbling block. Please elaborate.
Conclusions section:
- Please make a distinction here between between 'alternative cores' and 'independent bioclades'.
- You seem to be saying that "the autotrophic core essentially originated in a natural version of the Urey-Miller experiment." More attention to Ref. 8 would seem appropriate here.
Round 2
Reviewer 1 Report
All of my concerns have been addressed.